# Simulation and Experiment of Sieving Process of Sieving Device for Tiger Nut Harvester

**Hongmei Zhang, Zheng Zhou, Zhe Qu, Zhijie Li and Wanzhang Wang \***

College of Mechanical and Electrical Engineering, Henan Agricultural University, Zhengzhou 450002, China
\* Correspondence: wangwz@henau.edu.cn

**Abstract:** In order to realize mechanized and efficient harvesting of tiger nuts, study the efficient screening technology of beans and soil in a mechanized harvesting operation and improve the harvesting operation efficiency of crawler-type tiger nut harvesters, a theoretical analysis of the motion process of detritus particles on a sieve surface was conducted to determine the main factors affecting the motion of the particles on the sieve surface. A numerical simulation of the sieving process using the discrete element method was conducted to improve the screening efficiency of tiger nuts. The transport law of the debris particle population was analyzed from different perspectives, such as the average velocity of particle motion, particle distribution rate, screening efficiency and loss rate. The effects of factors such as screen amplitude (*SA*), vibration frequency (*VF*) and inclination angle (*IA*) on the sieving performance of the tiger nut threshing and screening device were investigated. The results show that sieving performance evaluation indexes, such as the average speed of particle movement, particle distribution rate, screening efficiency and loss rate, are non-linearly related to the factors of screen amplitude, vibration frequency and screen inclination angle; the effects of amplitude and frequency on the distribution particle size are consistent and show a gradual increase, with the distribution particle size reaching 3.32 mm at an amplitude of 14 mm and 3.46 mm at a frequency of 22 Hz. In the sieving process, the average velocity of the particle population decreases gradually along the direction of motion, and the influence of each factor on the average velocity of the particle population in the motion of the detritus is similar, all showing an increasing trend. This study can provide a reference for exploring the transport law of particles and the efficient screening technology of tiger nuts. Field harvesting tests showed that the screening efficiency and loss rate were 92.87% and 0.83%, respectively, at a screen amplitude of 14 mm, a vibration frequency of 10 Hz and an inclination angle of 2°, and the test results corresponded to the simulation results and met the design requirements of the tiger nut harvester. This study can provide reference for the investigation of the particle transport law and efficient screening technology for tiger nuts.

**Keywords:** tiger nuts; threshing and sieving device; movement of the particles; DEM

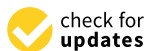

## 1. Introduction

The tiger nut was introduced to China mainly because it is a highly nutritious oil crop that can be used to ease the tight supply of cooking oil in the country [1,2]. The main reason why tiger nuts have not been developed on a large scale since their introduction into China in 1953 is attributed to harvesting problems [3]; therefore, in order to achieve mechanized harvesting of tiger nuts, problems such as high soil content and difficulties in separating beans and soil during harvesting should be solved. Vibratory screening devices are a core part of crop harvesting machinery [4–7] and numerous scholars have examined the screening performance through numerical simulations or prototype tests [8–14]. The discrete element method (DEM) is a numerical simulation method specifically designed to solve discontinuous media problems, which can effectively predict the macroscopic motion of particle populations [15–18]. In order to carry out the research of sieving theory and sieving mechanisms more efficiently, scholars at home and abroad have conducted a

great deal of research in combination with discrete element techniques [19–21]. Among them, Tijskens [22], Li [23], and Li [24,25] studied the motion of particles during sieving and revealed the decisive role of discrete mass motion on the screening efficiency. Both the difference in the structure of the screening device and the variation in the operating parameters can affect the screening performance by changing the movement, stratification and permeability of the debris particles. Tung [26], Wang [27], Wan [28], Zhang [29] and Wang [30] studied the influence of the structural and motion parameters of the vibrating screen on the screening efficiency. Chen [31], Zhang [32] and Li [33] addressed the problem of difficulty in grasping the distribution particle size in the process of particle sieving and found the influencing factors of sieving distribution particle size by using discrete element simulation analysis.

In order to improve the mechanized harvesting efficiency of tiger nuts, the group designed a tracked self-propelled tiger nut harvester based on the harvesting concept of "threshing first and separating later". On the basis of the previous research, this paper takes the beans and soil particle population as the research object and explores the motion characteristics of the debris particle population on the double-layer anisotropic reciprocating vibrating screen surface based on the motion state of the debris particle population simulated by EDEM. The change of the average velocity of the detritus particles was obtained and the transport law of the detritus particle population was analyzed. Combined with the distribution curve of the detritus particles, the mechanism of the influence of various factors on the sieving performance was studied with the screening efficiency and loss rate as indicators. It provides a reference for the design of tiger nut harvesting machinery.

## 2. Structure and Working Principle of Screening Device

### 2.1. Structure of Screening Device

The self-propelled tiger nut harvester mainly includes a digging device, lifting device, sieving device, driving device and grain bin (Figure 1). Among them, the sieving device is the core part of the tiger nut harvester, and its performance directly affects the efficiency of the harvester. The model of sieving device is shown in Figure 2, which mainly consists of a threshing drum, a double-layer anisotropic vibrating screen and a soil crushing guide roller.

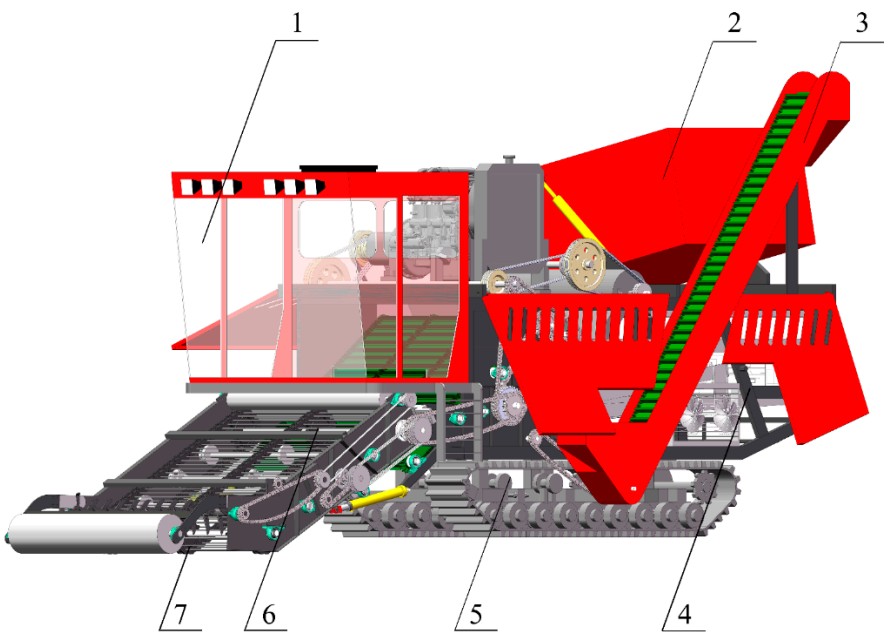

**Figure 1.** Self-propelled harvester of tiger nuts. 1. Cab 2. Grain bin 3. Squeegee lift conveyer 4. Sieving device 5. Crawler traveling device 6. Secondary lifting device 7. Digging device.

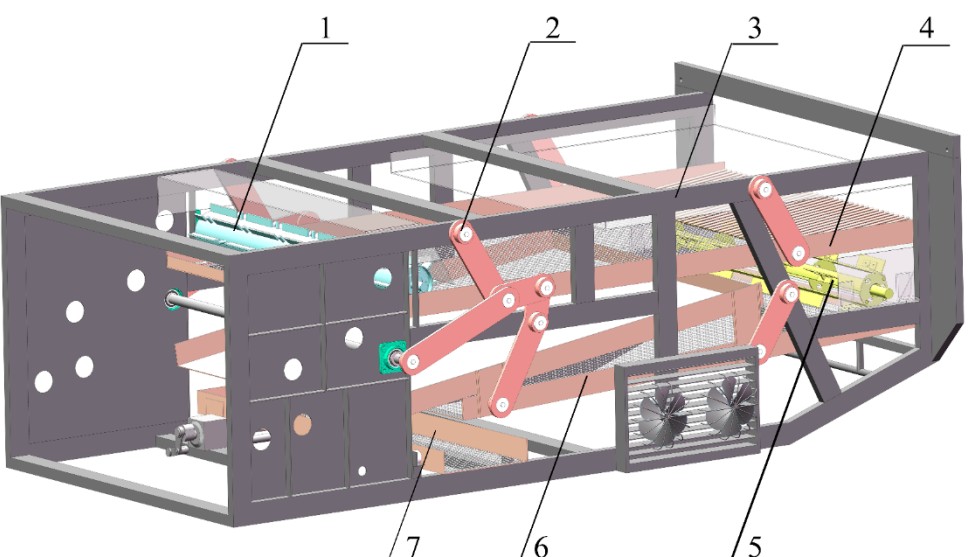

**Figure 2.** 3D model of the sieving device. 1. Tangential flow threshing drum 2. Linkage 3. Frame 4. Upper sieve 5. Guide drum 6. Lower sieve 7. Horizontal sieve.

*2.2. Working Principle*

As shown in Figure 3, the mixture of tiger nuts, soil and grass is fed into the threshing chamber after passing through the digging and lifting system. The threshing drums in the threshing chamber are cross-arranged with the plate and column tooth threshing elements, and the tiger nuts are separated from the rootstock by the beating, kneading and squeezing action of the threshing elements. The mixture is uniformly discharged to the upper woven sieve for primary screening, where most of the soil particles pass through the sieve, fall onto the baffle and are discharged from the front of the baffle; the roots and stems move to the rear end of the sieve surface and are discharged directly from the harvester. The tiger nuts and the remaining part of the soil particles fall from the rear end of the upper screen plate to the lower woven screen by the action of the soil-crushing guide roller. The soil particles fall to the ground through the lower sieve, while the tiger nuts fall from the front of the lower woven sieve to the lateral conveyor sieve, where they are transported to the grain bin via the grain-lifting device.

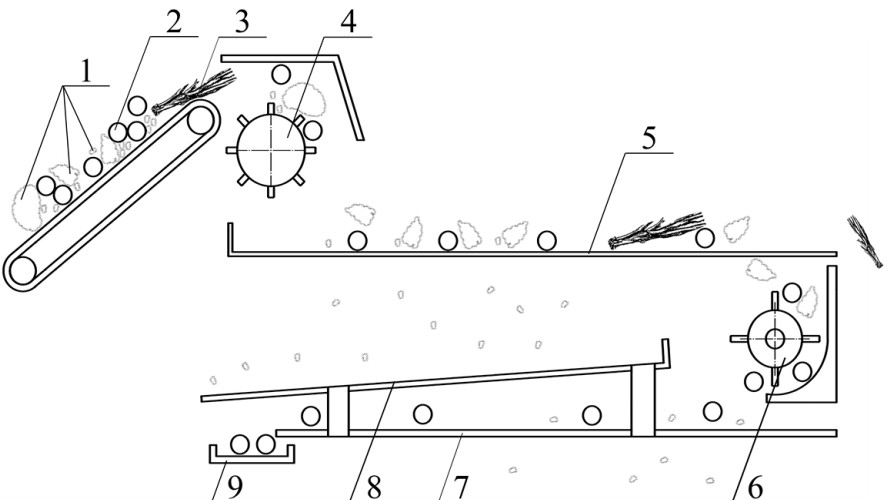

**Figure 3.** Schematic of the working principle. 1. Soil particles; 2. Tiger nut particles; 3. Stalk; 4. Threshing drum; 5. Upper sieve; 6. Guide drum; 7. Lower sieve; 8. retaining plate; 9. Horizontal sieve.

### 3. Analysis of the Process of Vibratory Screening of Exudate Particles

*3.1. Movement Analysis of Tiger Nut*

When the crank moves to the left half circumference, as shown in Figure 4a, $\cos \omega t < 0$, $Fg < 0$, the tendency of particle motion is to the left, and the direction of friction force points to the positive direction of x-axis; when the crank moves to the right half circumference, as shown in Figure 4b, $\cos \omega t > 0$, $Fg > 0$, the tendency of particle motion is to the right, and the direction of friction force points to the negative direction of x-axis.

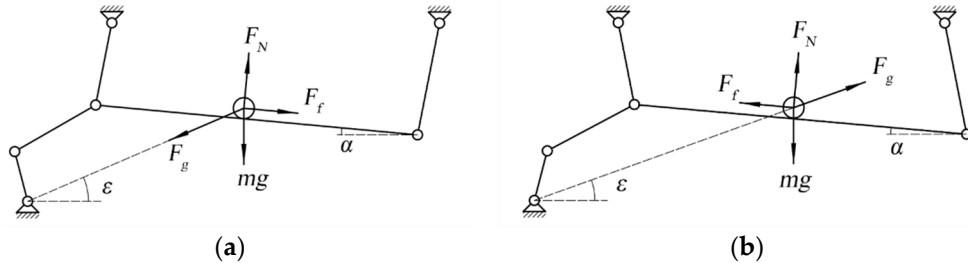

(**a**)　　　　　　　　　　　　　　　　　(**b**)

**Figure 4.** Stress analysis of tiger nut.

When the material particles produce slip along the sieve surface upward, and no leap, the force analysis of the particles is obtained:

$$F_N - mg \cos \alpha + \omega^2 \cos \omega t \sin(\varepsilon + \alpha) = 0 \tag{1}$$

$$ma_s = mR\omega^2 \cos \omega t \cos(\varepsilon + \alpha) - mg \sin \alpha - F_N \tan \varphi < 0 \tag{2}$$

where, $F_N$ is the support force of the particle; $m$ is the mass of the particle; $g$ is gravitational acceleration; $\varepsilon$ is the angle of vibration direction; $\alpha$ is the angle of inclination; $\varphi$ is the angle of friction between the particle and the sieve surface, $a_s$ is acceleration of the particle, m/s$^2$.

As shown in Equation (3), when the crank moves to the left half circumference and $R\omega^2/g > K_1$, the material particles move to the front end of the screen; the more the value of $R\omega^2/g$ is larger than the value of $K_1$, the greater the displacement of the particles moving to the front end.

$$\frac{R\omega^2}{g} > \frac{\sin(\alpha + \varphi)}{\cos(\varepsilon + \alpha - \varphi)} = K_1 \tag{3}$$

As shown in Equation (4), when the crank moves to the right half circumference and $R\omega^2/g > K_2$, the material particles move toward the back end of the screen surface; the more the value of $R\omega^2/g$ is larger than the value of $K_2$, the larger the displacement of particle movement.

$$\frac{R\omega^2}{g} > \frac{\sin(\varphi - \alpha)}{\cos(\varepsilon + \alpha - \varphi)} = K_2 \tag{4}$$

$$\frac{R\omega^2}{g} > \frac{\cos \alpha}{sin(\varepsilon + \alpha)} = K_3 \tag{5}$$

As shown in Equation (5), in the process of material particles moving toward the back end of the screen surface, when $R\omega^2/g > K_3$, the material is thrown off the screen surface. It can be seen that with the change of the motion parameters of the sieving device, the individual detachment particles present different motion states on the sieve surface and a large number of particles collide with each other on the sieve surface, leading to a strong randomness of particle motion; therefore, it is difficult to analyze the sieving process of the particle group using the theory, and thus, the following uses the discrete element method to simulate the motion process of the detachment particle group on the sieve surface and investigate the sieving mechanism of the detachment particles.

### 3.2. Modeling of Particles and Sieving Devices

A single spherical particle model was used to represent the bean particles, soil particles and stalks, respectively, using the variety "Spanish Garden Grain" as a reference (Figure 5). In order to ensure the accuracy of the particle model, the radius of soil particles is set in the range of 0.1 mm~3.9 mm in 0.1 mm increments, with a total of 39 size levels. The radius size range of the bean particles was set to 4 mm~4.5 mm, 5 mm, 6 mm, and 7 mm, for a total of nine size classes. The stalks were filled with round pellets of 2 mm radius, and the length of the filled model was 100 mm, which was consistent with the length of the stalks remaining in the field at the time of actual harvest.

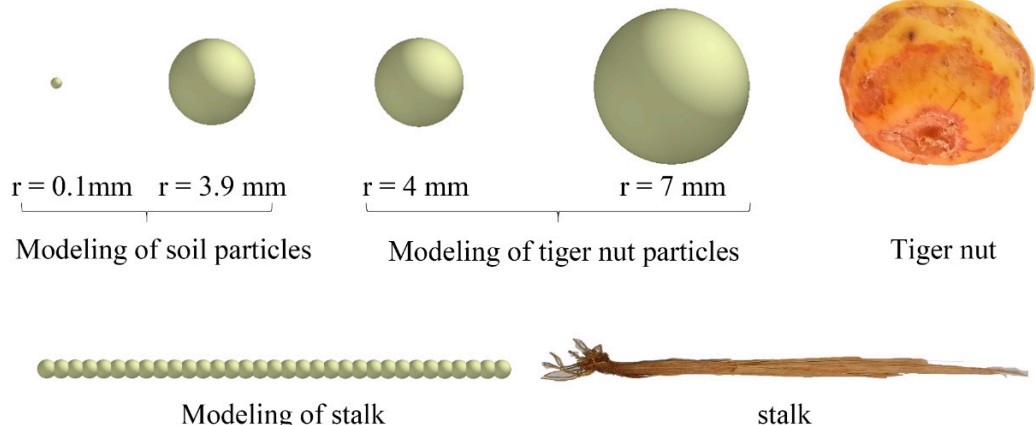

**Figure 5.** Models of tiger nut, soil particles and stalk.

The model used a roller diameter of 250 mm and length of 1620 mm, upper sieve length of 3000 mm and width of 1650 mm; crushing guide roller diameter of 280 mm and length of 1630 mm, lower sieve length of 2450 mm and width of 1660 mm. All woven sieves are square sieves with 8 mm side lengths. The 3D model is imported into the EDEM software, as shown in Figure 6. Since in the simulation of the sieving process of the debris particle population, the focus is on the simulation and analysis of the particle motion and the sieve penetration process, the Hertz–Mindlin (no-slip) model is chosen for the numerical simulation process, assuming that the recovery coefficient and collision intensity of the debris particles do not change, and the effect of air resistance on the debris particles is ignored. According to the research results of the group and relevant references, the mechanical and contact parameters of the selected materials are shown in Table 1 [34–36].

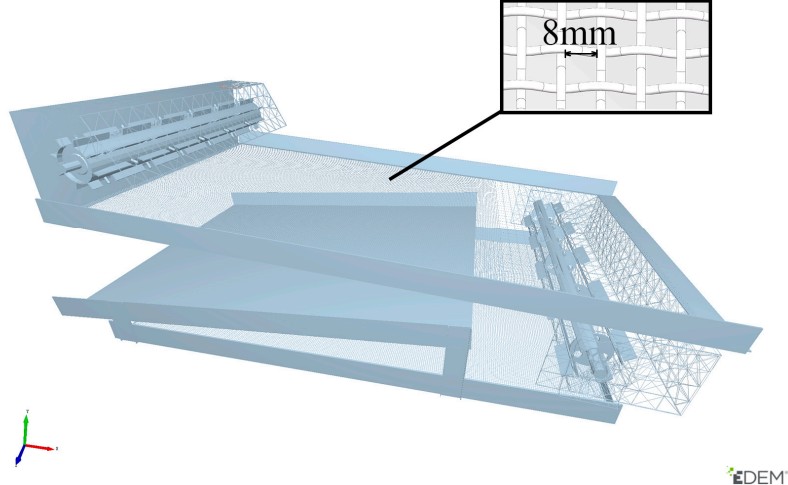

**Figure 6.** 3D model of the device inside the EDEM.

**Table 1.** Mechanical parameters and contact coefficients of materials.

| | Materials | Poisson Ratio | Density (kg/m³) | Shear Modulus (MPa) |
|---|---|---|---|---|
| Mechanical Parameters | Tiger nut | 0.18 | 1230 | 4 |
| | Soil particle | 0.26 | 1179 | 1.1 |
| | Steel | 0.27 | 7850 | $8 \times 10^4$ |
| | Stalk | 0.42 | 241 | 1.0 |
| | **Materials** | **Collision recovery coefficient** | **Static friction coefficient** | **Dynamic friction coefficient** |
| Contact coefficients | Tiger nut—Tiger nut | 0.48 | 0.10 | 0.34 |
| | Tiger nut—Soil particle | 0.49 | 0.25 | 0.42 |
| | Tiger nut—Steel | 0.62 | 0.07 | 0.25 |
| | Tiger nut—Stalk | 0.35 | 0.02 | 0.32 |
| | Soil particle—Soil particle | 0.14 | 0.27 | 0.56 |
| | Soil particle—Steel | 0.15 | 0.36 | 0.50 |
| | Soil particle—Stalk | 0.11 | 0.09 | 0.21 |
| | Stalk—Stalk | 0.26 | 0.01 | 0.32 |
| | Stalk—Steel | 0.43 | 0.03 | 0.45 |

During the particle swarm motion simulation, the computational domain dimensions are set to X [−2000, 2000], Y [−900, 900] and Z [−1000, 1000], and the gravitational acceleration is taken as 9.81 m/s². The total pellet feeding rate was determined to be 50 kg/s based on field yield measurements and harvesting requirements, with 37.5 kg/s of soil pellets, 10 kg/s of bean pellets and 2.5 kg/s of stalks. After field sampling, the mass fraction of particle size within a radius of 4 mm was measured using a soil sieve. The soil particle size was divided into eight groups and the tiger nut particle size was divided into four groups, and the mass fractions of different particle sizes within the EDEM software were set as shown in Table 2.

**Table 2.** Mass fraction of particles.

| Type of Particles | Range of Particle Sizes (mm) | Mass Fraction (%) |
|---|---|---|
| Soil particles | 0.1~0.5 | 7 |
| | 0.6~1 | 10 |
| | 1.1~1.5 | 17 |
| | 1.6~2 | 21 |
| | 2.1~2.5 | 18 |
| | 2.6~3 | 14 |
| | 3.1~3.5 | 9 |
| | 3.6~3.9 | 4 |
| Tiger nut particles | 4~4.5 | 26 |
| | 5 | 45 |
| | 6 | 19 |
| | 7 | 10 |

Based on the previous research results [37,38], in the simulation process, the screen amplitude (*SA*), vibration frequency (*VF*) and inclination angle (*IA*) are used as the setting variables. The values of amplitude of the screen surface are 6 mm, 8 mm, 10 mm, 12 mm and 14 mm; the values of vibration frequency are 14 Hz, 16 Hz, 18 Hz, 20 Hz and 22 Hz; and the values of inclination angle of the screen surface are 2°, 4°, 6°, 8° and 10°.

Throughout the entire screening process, the particle group not only have the collisions between the internal particles and the collision impacts produced by the vibrating screen, the particles in the process of movement to complete the dispersion and separation must also pass small particles through the screen. With the continuous feeding of the mixture, a large number of particles in the front end of the screen produce a buildup. As the sieving continues, the dynamic balance between the accumulation of particles on the screen surface

and the permeable screen will be reached, and finally, the sieving reaches a stable state, and the whole process is shown in Figure 7.

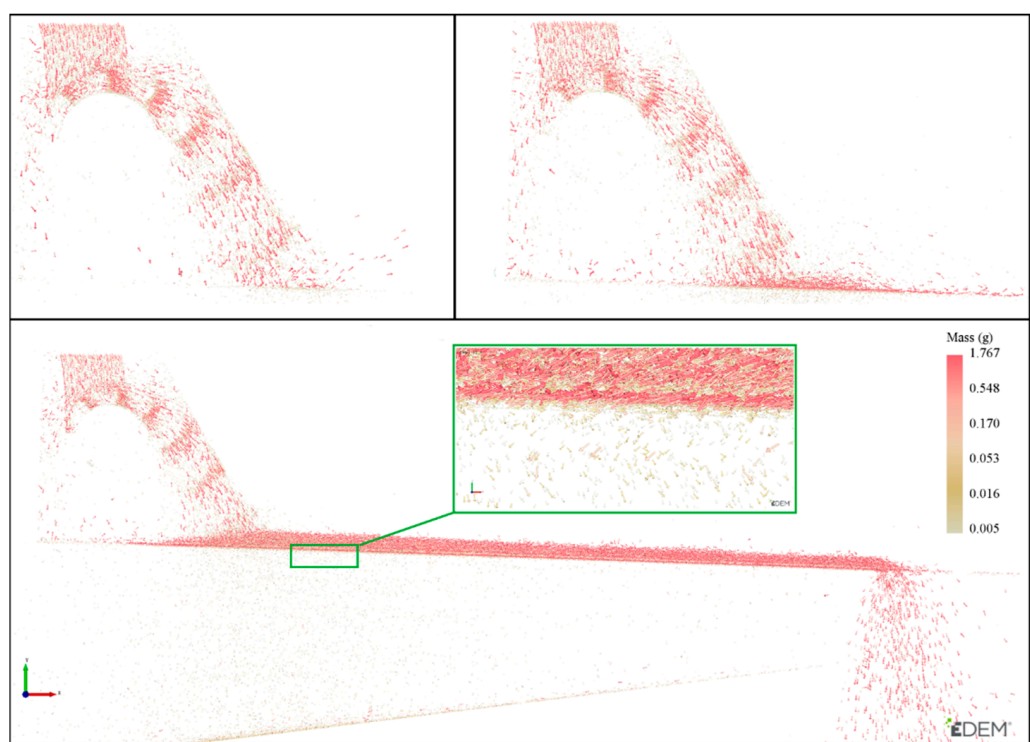

**Figure 7.** Movement of the particles during the sieving process.

*3.3. Effect of Factors on the Average Velocity of the Particle Population on the Screen*

The average speed of the particle group on the screen surface reflects the mobility of the particle group; the higher the average speed, the better the mobility of the particle group. The smaller particles are more likely to contact with the screen surface, improve the probability of particle penetration, and then enhance the screening efficiency; conversely, the screening efficiency is reduced. As can be seen from Figure 8, the velocity of the detritus particles after passing through the drum decreases significantly after contacting the screen surface, and the velocity gradually reaches a stable value during the movement with the screen surface. The average velocity of the particles all increases with the increase of screen amplitude, vibration frequency and inclination angle.

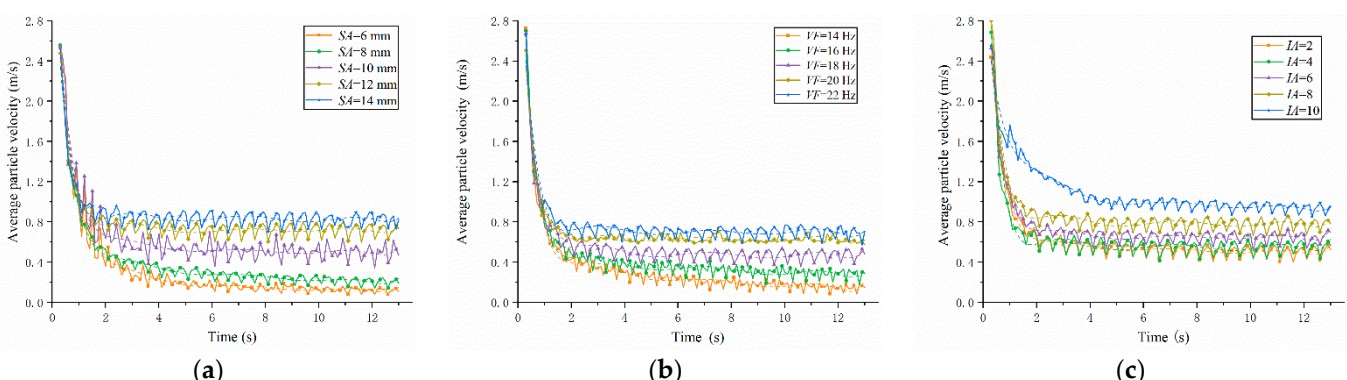

**Figure 8.** Curve of the average velocity of the particles. (**a**) Curve of the average velocity of the particles with amplitude; (**b**) Curve of the average velocity of the particles with frequency; (**c**) Curve of the average velocity of the particles with angle.

From the center fit line in Figure 8a, it can be seen that the average velocity of particles reaches a stable value after 2 s when *SA* is 14 mm, and the time for the average velocity of particles to reach a stable value is gradually pushed back as *SA* decreases. When *SA* is 6 mm, the average velocity of particles tends to a stable value only after 5 s. With the change of amplitude, the maximum average velocity of particles on the screen surface fluctuates around 0.82 m/s, while the minimum average velocity fluctuates around 0.12 m/s, with a difference of 0.7 m/s between the maximum average velocity and the minimum average velocity. From Figure 8b, it can be seen that the average particle velocity increases from 0.16 m/s to 0.7 m/s when the *VF* increases from 14 Hz to 22 Hz; when the frequency is 14 Hz, the average particle velocity reaches a value that tends to be stable only after 9 s, which indicates that the mobility of the detritus particles on the screen surface is poor, making it is easy to cause accumulation on the screen surface. Figure 8c shows that when *IA* is 2°, the average velocity of particles is 0.5 m/s. As the inclination angle of the screen surface increases, the average velocity of particles at the screen surface gradually increases, but it does not show a linear increasing trend, and the amount of change of the average velocity is 0.04 m/s, 0.11 m/s, 0.13 m/s and 0.17 m/s in order from small to large. By comparing Figure 6, it can be obtained that when the amplitude, frequency and inclination angle all increase by the same value, the average particle velocity varies the most with the amplitude, reaching 0.7 m/s, while the average particle velocity varies the least with the inclination angle, with a value of 0.44 m/s.

### 3.4. Effect of Factors on Distribution Particle Size

The distribution of particle size is an important indicator of sieving performance, and regardless of the structure of the sieving device, the sieving work is done with a certain distribution of particle size. In this device, the edge length of the woven sieve is 8 mm, but in the actual sieving process, not all particles with radius less than 4 mm can pass through the sieve, so it is important to study the particle size distribution under different parameters to explore the efficient harvesting of materials. The distribution size of the particles on the screen (r50) is the particle radius corresponding to a distribution rate ($\zeta$) equal to 50%. The partition rate is calculated by the following equation.

$$\zeta = \frac{n_1}{n_2} \times 100\% \tag{6}$$

where, $n_1$ is the number of particles on the sieve surface, $n_2$ is the total number of particles.

$S_1$ and $S_2$ in Figure 9 indicate the particles that should be permeable but are not permeable to the sieve and the particles that should not be permeable but are permeable to the sieve, respectively. According to the changes of the distribution rate curve in Figure 9a, it can be seen that with the gradual increase of the amplitude, the distribution rate curve moves to the right side as a whole, and the corresponding $S_1$ gradually decreases, indicating that the number of small particles on the screen surface gradually decreases and the sieving performance gradually improves. At an amplitude of 14 mm, the particle size distribution reaches a maximum of 3.3 mm, which is 0.8 mm higher than the distribution size corresponding to an amplitude of 6 mm. The effect of frequency on the partition size can be obtained from the plot of Figure 9b, where the partition rate curve shifts to the right as the frequency increases from 14 Hz to 22 Hz, the partition size increases from 2.9 mm to 3.5 mm, and the number of small particles passing through the sieve increases. Figure 9c shows the variation of the distribution rate curve with the inclination angle; at the inclination angle of 6°, the curve corresponds to the minimum value of $S_1$, while the values of $S_1$ and $S_2$ are the closest, indicating the best comprehensive screening efficiency at this time. When the inclination angle is 2°, 4°, 6°, 8° or 10°, the corresponding distribution particle sizes are 3.1 mm, 3.3 mm, 3.4 mm, 3.3 mm or 3.2 mm, respectively. The distribution size tends to increase and then decrease with the increase of inclination angle. It can be seen that the increase of amplitude and frequency can effectively promote the stratification situation of small particles on the screen surface, increase the contact probability between

small particles and the screen surface, and improve the distribution particle size of particles on the screen surface. The influence of the inclination angle on the distribution of particle size is first increased and then reduced, mainly because the appropriate increase in the inclination angle of the screen surface can promote the movement of the particles on the screen surface, small particles are more likely to contact with the screen surface, improve the probability of sieve penetration. If the inclination angle is too large, the movement of the particles is too fast, producing a result in which the particles cannot effectively make contact with the screen surface, thus reducing the probability of the particles penetrating the screen.

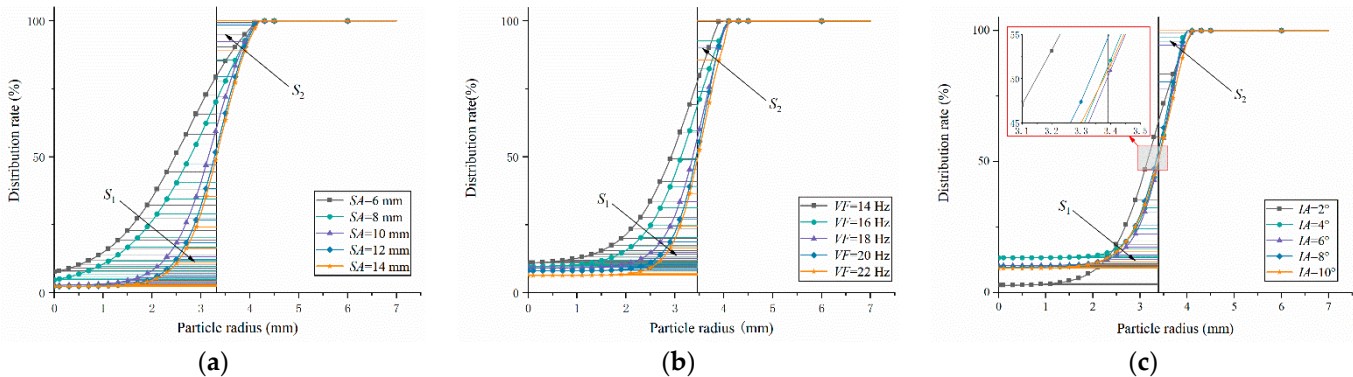

**Figure 9.** Curve of the distribution rate. (**a**) Curve of the distribution rate with amplitude; (**b**) Curve of the distribution rate with frequency; (**c**) Curve of the distribution rate with angle.

*3.5. Effect of Various Factors on Screening Efficiency and Loss Rate*

The screening efficiency and loss rate give the best visual indication of the sieving performance of the sieving unit. The screening efficiency ($Y_1$) is the ratio of the mass of soil particles under the sieve to the total mass of soil particles, and the loss rate ($Y_2$) is the ratio of the mass of lost tiger nuts to the total mass of tiger nuts.

From Figure 10a, it can be seen that the screening efficiency and loss rate both increase with the increase of amplitude, and the amount of change in screening efficiency reaches 38.2% during the increase of amplitude from 6 mm to 10 mm, and the degree of change in screening efficiency is smaller when the amplitude increases from 10 mm to 14 mm, which only increases from 88.46% to 93.53%. Throughout the variation, the loss rate increased from 0.57% to 5.47%, and it is noteworthy that the loss rate increased by 0.16% when the amplitude increased from 10 mm to 12 mm, and by 0.97% when the amplitude increased from 12 mm to 14 mm. According to Figure 10b, the screening efficiency and loss rate increased with the increase of frequency in the range of vibration frequency, and the amount of change of screening efficiency reached 31.47% and the amount of change of loss rate reached 5.7%. In the frequency increase from 14 Hz to 16 Hz, the screening efficiency changes most significantly from 61.52% to 77.45%, and the screening efficiency reaches a maximum value of 92.99% at a frequency of 22 Hz. The loss rate increased by 3.5% when the frequency was increased from 14 Hz to 18 Hz and by 2.2% when the frequency was increased from 18 Hz to 22 Hz. The effect of screen surface amplitude and sieving frequency on screening efficiency and loss rate is consistent, with the increase of the value of the test factor, the screening efficiency and loss gradually increased, the reason is that the increase of amplitude and frequency promotes the movement, separation and dispersion of particles in horizontal and vertical directions, which improves the probability of particle penetration and promotes the screening efficiency. At the same time, the violent movement of particles leads to the increase of loss rate. Figure 10c shows that in the process of increasing the inclination angle from 2° to 10°, the screening efficiency showed a trend of increasing and then decreasing, and the screening efficiency reached the maximum value of 93.84% at the inclination angle of 6°. When the inclination angle increased from 2° to 6°, the screening efficiency increased by 5.38%, and when the inclination angle increased from 6° to 8°, the

screening efficiency decreased by 12.46%. In the range of inclination angle change, the loss rate showed a gradually increasing trend from 4.34% to 6.48%. The increase of the inclination angle improves the movement of the particles on the screen surface and the dispersion of the particles is enhanced, which is conducive to improving the screening efficiency; too large of an inclination will cause the particles to move too fast and reduce the number of collision contacts with the screen surface, reducing the screening efficiency.

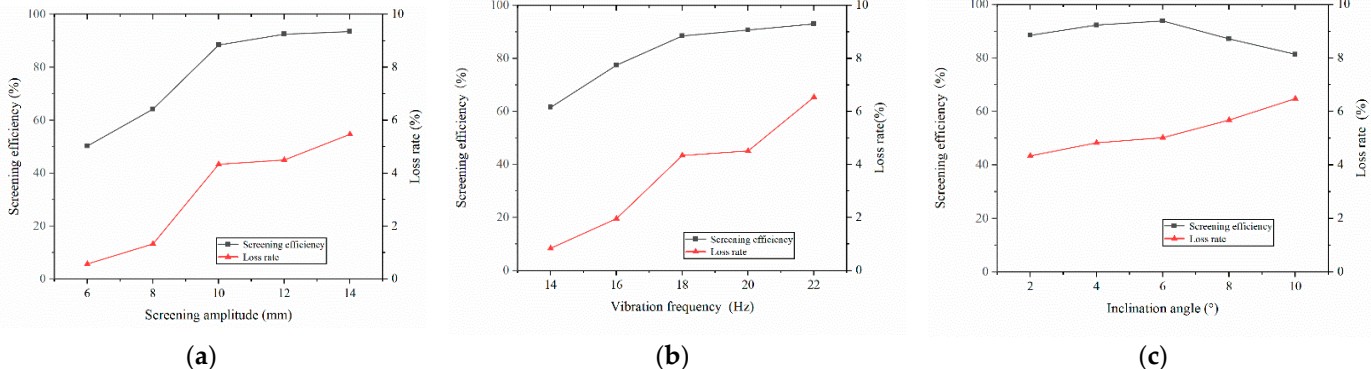

**(a)**　　　　　　　**(b)**　　　　　　　**(c)**

**Figure 10.** Curve of the screening efficiency rate and loss rate. (**a**) Curve of the screening efficiency rate and loss rate with amplitude; (**b**) Curve of the screening efficiency rate and loss rate with frequency; (**c**) Curve of the screening efficiency rate and loss rate with angle.

*3.6. Impact of Interaction Terms and Optimization of Multiple Parameters*

Screening efficiency and loss rate during harvesting are influenced by the interactions between various factors. A total of 27 groups of experiments were designed. The test factor codes are shown in Table 3 and the test results are shown in Table 4.

**Table 3.** Codes of experimental factors.

| Codes | Factors | | |
|:---:|:---:|:---:|:---:|
| | $X_1$ (Hz) Vibration Frequency | $X_2$ (mm) Screen Amplitude | $X_3$ (°) Inclination Angle |
| −1 | 14 | 6 | 2 |
| −0.5 | 16 | 8 | 4 |
| 0 | 18 | 10 | 6 |
| 0.5 | 20 | 12 | 8 |
| 1 | 22 | 14 | 10 |

The mathematical models of screening efficiency and loss rate were established by ANOVA of the screening efficiency and loss rate in the test results, as shown in Equations (7) and (8), respectively. The order of significance of the three main effects on the screening efficiency is inclination angle, screen amplitude and vibration frequency, in descending order. The response surface for the effect of the interaction of the factors on the screening efficiency is shown in Figure 11. The order of significance of the three main effects on the loss rate is screen amplitude, vibration frequency and inclination angle, in descending order of significance. The response surface for the effect of the interaction of the factors on the loss rate is shown in Figure 12.

$$
\begin{aligned}
Y_1 = {} & 94.24 + 0.56X_1 + 1.48X_2 + 1.73X_3 \\
& + 1.32X_1X_2 - 2.65X_1X_3 - 3.23X_2X_3 \\
& + 1.36X_1{}^2 - 0.65X_2{}^2 + 1.39X_3{}^2
\end{aligned}
\tag{7}
$$

$$
\begin{aligned}
Y_2 = {} & 3.17 + 1.27X_1 + 2.08X_2 + 0.73X_3 \\
& + 1.4X_1X_2 - 0.94X_1X_3 - 1.09X_2X_3 \\
& + 0.8X_1{}^2 + 0.46X_2{}^2 - 0.074X_3{}^2
\end{aligned}
\tag{8}
$$

**Table 4.** Codes of experimental factors.

| No. | Factors | | | Evaluation Index | |
|---|---|---|---|---|---|
| | $X_1$ (Hz) Vibration Frequency | $X_2$ (mm) Screen Amplitude | $X_3$ (°) Inclination Angle | $Y_1$ (%) Screening Efficiency Rate | $Y_2$ (%) Loss Rate |
| 1 | −1 | −1 | −1 | 88.69 | 0.18 |
| 2 | −1 | −0.5 | −0.5 | 91.53 | 0.61 |
| 3 | −1 | 0 | 0 | 95.30 | 2.79 |
| 4 | −1 | 0.5 | 0.5 | 96.92 | 4.06 |
| 5 | −1 | 1 | 1 | 96.73 | 4.42 |
| 6 | −0.5 | −1 | −0.5 | 89.15 | 0.59 |
| 7 | −0.5 | −0.5 | 0 | 93.80 | 1.97 |
| 8 | −0.5 | 0 | 0.5 | 96.72 | 3.28 |
| 9 | −0.5 | 0.5 | 1 | 97.15 | 3.88 |
| 10 | −0.5 | 1 | −1 | 96.08 | 3.92 |
| 11 | 0 | −1 | 0 | 91.89 | 1.71 |
| 12 | 0 | −0.5 | 0.5 | 95.70 | 2.94 |
| 13 | 0 | 0 | 1 | 97.37 | 3.94 |
| 14 | 0 | 0.5 | −1 | 95.80 | 4.23 |
| 15 | 0 | 1 | −0.5 | 96.69 | 6.17 |
| 16 | 0.5 | −1 | 0.5 | 94.69 | 1.90 |
| 17 | 0.5 | −0.5 | 1 | 97.13 | 3.32 |
| 18 | 0.5 | 0 | −1 | 95.47 | 3.75 |
| 19 | 0.5 | 0.5 | −0.5 | 96.60 | 6.03 |
| 20 | 0.5 | 1 | 0 | 96.26 | 7.30 |
| 21 | 1 | −1 | 1 | 96.02 | 3.44 |
| 22 | 1 | −0.5 | −1 | 96.03 | 2.80 |
| 23 | 1 | 0 | −0.5 | 96.56 | 5.51 |
| 24 | 1 | 0.5 | 0 | 97.18 | 7.26 |
| 25 | 1 | 1 | 0.5 | 96.79 | 8.09 |
| 26 | 0 | 0 | 0 | 94.24 | 3.31 |
| 27 | 0 | 0 | 0 | 94.65 | 2.95 |

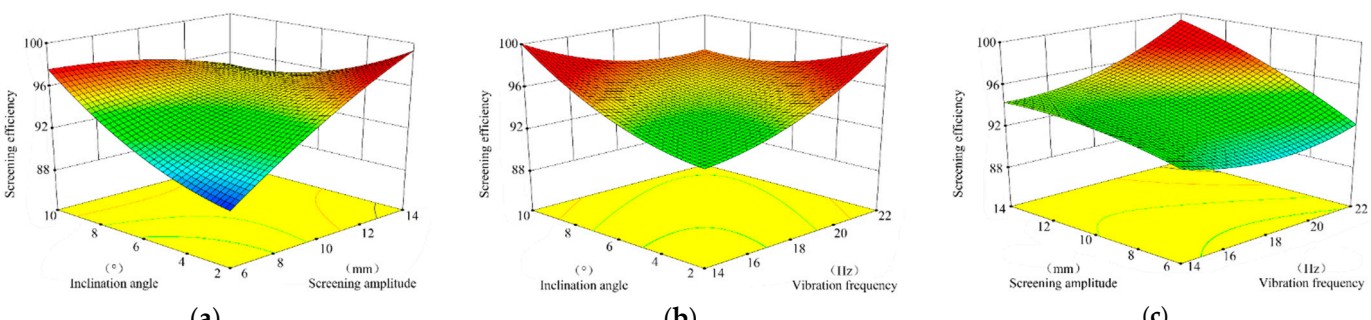

(**a**)       (**b**)       (**c**)

**Figure 11.** The response surface for the effect of the interaction of the factors on the loss rate. (**a**) Response surface of the effect of *SA* and *IA* on screening efficiency; (**b**) Response surface of the effect of *VF* and *IA* on screening efficiency; (**c**) Response surface of the effect of *VF* and *SA* on screening efficiency.

According to the established mathematical model, the factors were optimized with the aim of finding the best working parameters. The optimization of the working parameters and the predicted values of the evaluation index are shown in Table 5.

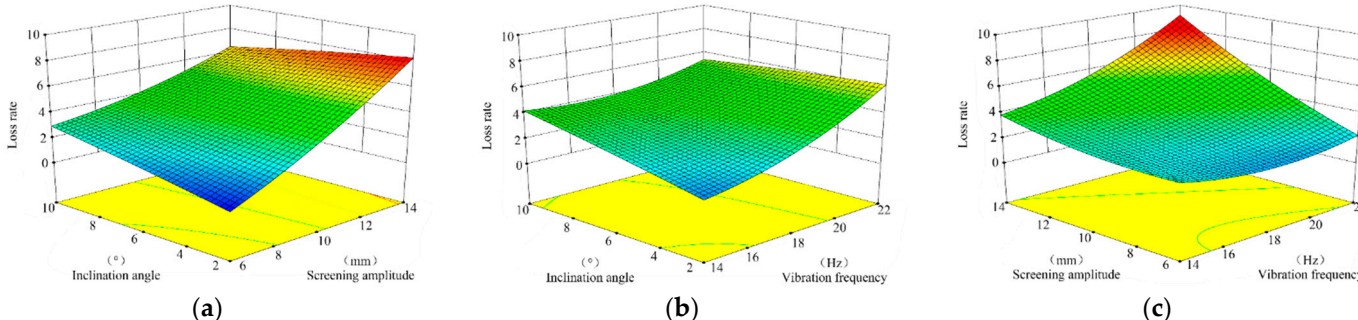

**Figure 12.** The response surface for the effect of the interaction of the factors on the screening efficiency. (**a**) Response surface of the effect of *SA* and *IA* on loss rate; (**b**) Response surface of the effect of *VF* and *IA* on loss rate; (**c**) Response surface of the effect of *VF* and *SA* on loss rate.

**Table 5.** Optimization of multiple parameters.

| Name | Goal | Lower Limit | Upper Limit | Lower Weight | Upper Weight | Importance | Selected |
|---|---|---|---|---|---|---|---|
| $X_1$ (Hz) | is in range | 14 | 22 | 1 | 1 | 3 | 14 |
| $X_2$ (mm) | is in range | 6 | 14 | 1 | 1 | 3 | 10 |
| $X_3$ (°) | is in range | 2 | 10 | 1 | 1 | 3 | 2 |
| $Y_1$ (%) | maximize | 88.6899 | 97.368 | 1 | 1 | 2 | 91.73 |
| $Y_2$ (%) | minimize | 0.180984 | 3 | 1 | 1 | 3 | 0.79 |

## 4. Field Trials

The field harvest trial was conducted at the trial field in Guodian Town, Xinzheng City, Henan Province, China, where the tiger nuts variety planted was Spanish round grain, the planting pattern was monopoly crop, and the soil moisture content at harvest was 11%, and the field harvest trial is shown in Figure 13. Before each test, the sieve surface and materials in the grain bin were cleaned. The travel speed of the tiger nut harvester was 1.8 km/h, the digging depth was 130 mm, the harvesting width was 1800 mm, and the working conditions of the tiger nuts harvester were kept constant during the test. The operating parameters of the sieving device during the harvesting operation of the tiger nut harvester were adjusted to a vibration frequency of 14 Hz, a screen amplitude of 10 mm and an inclination angle of 2°. According to the standard of tiger nut harvesting, screening efficiency and loss rate were selected as performance indicators. The screening efficiency ($\eta$) was

$$\eta = \frac{m - m_t - m_s}{m - m_t} \times 100\% \tag{9}$$

where, $m$ is the total mass of the fed material, $m_t$ is the mass of tiger nuts in the grain bin and $m_s$ is the mass of soil in the grain bin.

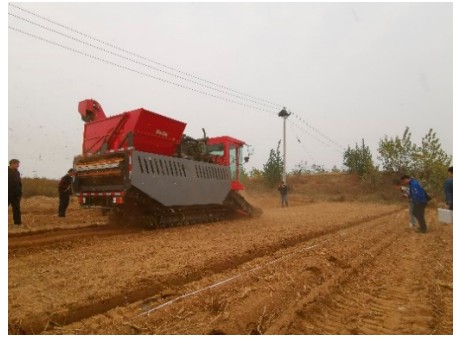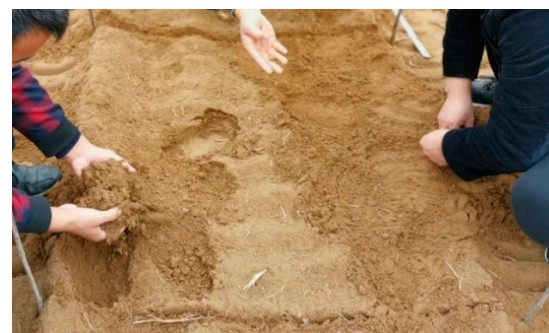

**Figure 13.** Trials for field harvesting.

The loss rate ($\psi$) is

$$\psi = \frac{m_l}{m_t + m_l} \times 100\% \tag{10}$$

where, $m_l$ is the mass of lost tiger nuts.

Based on the results of the previous analysis, three replicated field trials were conducted under the conditions of the preferred operating parameters of the threshing and screening device, and the results are shown in Table 6. Field test results show that the average value of screening efficiency is 92.87% and the average value of loss rate is 0.83% under the working conditions of 10 mm screen amplitude, 14 Hz vibration frequency and 2° inclination angle of the screen surface. The standard deviations were 1.13% and 4.2%, indicating that the results of the three data sets were real and reliable, without outliers. By comparing with the simulation test results, it can be seen that the error of screening efficiency in the simulation test results is 1.23% and the error of loss rate is 4.82%, which are less than 5%, indicating that the simulation results have certain reliability and that the results meet the design requirements of the tiger nut harvester.

**Table 6.** Results of field trials.

| No. | Screening Efficiency Rate (%) | Loss Rate (%) |
| --- | --- | --- |
| 1 | 91.98 | 0.77 |
| 2 | 93.52 | 0.87 |
| 3 | 93.11 | 0.84 |
| Average | 92.87 | 0.83 |
| Standard deviation | 1.13 | 4.2 |
| Error of simulation results | 1.23 | 4.82 |

## 5. Discussion

The tiger nut industry is in its infancy in China, and a solution to the harvesting problem is urgently needed. Solving the contradictory problems of high efficiency and low loss is the focus of this study. In this paper, a sieving device for a self-propelled tracked bean harvester is designed to address the problems of difficult manual harvesting, low mechanized harvesting efficiency and high loss rate of tiger nuts. The structure of this device is somewhat innovative; the rollers are effective in reducing the entrapment losses caused by the encapsulation of tiger nuts by the rootstock and the design of double-deck reciprocating vibrating screen provides tiger nuts the maximum screening stroke in the limited body space, thereby increasing the effective screening area and improving the screening efficiency.

In this paper, through theoretical analysis, simulation and field test, we apply discrete element technology to simulate the motion process of particles on the sieve surface, and reveal the motion characteristics of tiger nuts on the sieve surface. The sieving process can be described as follows: if the particles can be thrown off the screen surface under the influence of the excitation force of the screen surface, the particles will be forced to vibrate accordingly with the movement of the screen surface; if the particles cannot be thrown off the screen surface under the influence of the excitation force of the screen surface, the particles will be more random in the subsequent movement, and all factors influence the migration of particles on the sieve surface, the higher the value, the more effective the migration of particles. From different perspectives, such as the average speed of particle movement, distribution rate of particles, screening efficiency and loss rate, the sieving mechanism of tiger nuts on the sieve surface is revealed, providing a theoretical basis and technical basis for the development of tiger nut harvesting equipment and the improvement of operational performance. At the same time, the methods and ideas in this paper are also suitable for other types of particle sieving and collection.

This paper presents an innovation in the structure of the tiger nut screening device, but there are still some limitations. Field trials have shown that the machine can adequately meet the design requirements and can significantly increase the harvesting efficiency of

tiger nuts, but a large number of field trials are needed in the future to further study the adaptability of this device to different varieties of tiger nuts and to collect a large amount of operational and field data in preparation for improving the intelligence of the harvester.

## 6. Conclusions

(1) Based on discrete element software, a model of soil particles and bean particles were established to clarify the effects of structural parameters and motion parameters (screen amplitude, vibration frequency and inclination angle) of the device on the bean and soil separation process and sieving performance.

(2) The simulation analysis reveals that the average velocity of the particle population gradually decreases along the direction of motion during the sieving process, and the influence of each factor on the average velocity during the motion of the particle population of the detritus is similar, all showing an increasing trend. The effects of amplitude and frequency on the distribution size are consistent, both showing a gradual increase, reaching 3.32 mm for an amplitude of 14 mm and 3.46 mm for a frequency of 22 Hz. The distribution particle size increases with the increase of the screen inclination angle and then decreases, and the distribution particle size reaches the maximum value of 3.39 mm when the screen inclination angle is 6°.

(3) The field harvesting test shows that the average values of screening efficiency and loss rate are 92.87% and 0.83%, respectively, at the sieve amplitude of 14 mm, vibration frequency of 10 Hz and inclination angle of 2°, which meet the design requirements of the tiger nut harvester. These results are basically consistent with the simulation test results, verifying the accuracy of the simulation results.

## 7. Patents

Qu, Z.; He, X.; Wang, W.Z.; Zhou, Z.; Lv, Y.L.; Guo, H.Q. Caterpillar Self-propelled Tiger Nut Harvester and Harvesting Method of Tiger Nut: ZL202011123909.8[P]. 11 December 2020.

**Author Contributions:** Conceptualization, H.Z. and W.W.; methodology, H.Z. and Z.Z.; investigation, Z.Q. and Z.Z.; Visualization, Z.Z.; writing—original draft preparation, Z.Z. and Z.L. All authors have read and agreed to the published version of the manuscript.

**Funding:** This research was funded by the special fund for National Key R&D Program of China (Grant No. 2019YFD1002602).

**Institutional Review Board Statement:** Not applicable.

**Data Availability Statement:** The data used to support the findings of this study are available from the corresponding author upon request.

**Acknowledgments:** The authors would like to thank their college and the laboratory, as well as gratefully appreciate the reviewers who provided helpful suggestions for this manuscript.

**Conflicts of Interest:** The authors declare no conflict of interest.

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
