# Peer review of "Simulation and Experiment of Sieving Process of Sieving Device for Tiger Nut Harvester"

_agriculture, doi:10.3390/agriculture12101680_

Round 1
Reviewer 1 Report
The authors of the work presented the analysis of the tiger nut harvesting machine, and more precisely the screening analysis during harvesting. The work is very interesting. It is wondering whether this machine can still be used for harvesting other crops. The analysis carried out by the authors is correct, but I would describe the simulation in more detail.
In addition, I would describe the possibility or justify it not using the machine for other harvesting.
The authors were not able to avoid editorial mistakes.
1) in lines 42, 44, 47 and 104 there should be a space after the period.
2) in lines 37, 40, 42, 44, 46, 47, 51, 52, 53 and 122 there should be a space between the word and the parenthesis with the source number.
3) in line 122 there should be another period at the end.
4) in line 126 there should be a space between the axis and the values in parentheses.
Author Response
Dear Professor:
I have studied the valuable comments from reviewers carefully, and tried our best to revise the manuscript. This machine can also harvest other root crops, more detailed simulation process has been added to the text. Tiger nut has relatively small volume, low manual harvesting rate, low harvesting efficiency and high cost, so it is not suitable for large area harvesting operations. In addition, 1-4 all formatting issues have been modified in the main text.

Reviewer 2 Report
Aiming at the difficulty of mechanized harvesting of tiger nuts, it is of practical significance to carry out simulation research on the vibration process of vibratory screening devices, key components of harvesters. But the manuscript still has the following shortcomings.
1.The title of the manuscript is somewhat different from its content. The content of the manuscript does not support the sieving process of sieving device in the title. The quantifiable data of particle speed, distribution particle size, screening efficiency and loss rate obtained through simulation are not enough. The interaction process between particles and particles-sieve on the sieve surface are lacking, and these can just reflect the sieving process of sieving device.
2.The manuscript lacks the theoretical analysis of the sieving process of sieving device and the determination of the experimental factors. Author does not explain why the screen amplitude (SA), vibration frequency (VF), inclination angle (IA) and their levels are selected as the experimental factors, which makes the analysis of the effect of the experimental factors not deep enough.
3.The test plan of the manuscript is too simple and needs to be optimized. Does the interaction of the screen amplitude, vibration frequency and inclination angle affect the test evaluation index?
4.The working parameters of the sieve surface in the field test and its realization method are not specified. There are too few field verification experiments. The optimal parameters in the simulation test cannot be proved to be the optimal parameters in practical application; and it is necessary to carry out the adaptability test of different varieties of tiger nut.
5.The discussion part of the manuscript does not meet the writing requirements of this part.
6.The manuscript lacks relevant pictures of the simulation process. It is recommended to combine these pictures to explain the impact analysis of experimental factors.
7.The data in the manuscript discussion is inconsistent with the previous text, which makes the data in the manuscript questionable.
8.Some characters and numbers are wrongly written, such as Shear modulus of Steel. “As can be seen from Figure 8,” should be “As can be seen from Figure”.
9.The relevant expressions in the manuscript should be consistent, such as tiger nut particles, oilseed beans, and oleaginous beans should be unified.
10.The references are almost entirely in Chinese, which is very bad.
11.Some references are formatted incorrectly, Shun as NO. 23.

Author Response
Dear Professor:
I have studied the valuable comments from reviewers carefully, and tried our best to revise the manuscript. The point to point responds to the reviewer’s comments are listed as following:
Comment 1: The title of the manuscript is somewhat different from its content. The content of the manuscript does not support the sieving process of sieving device in the title. The quantifiable data of particle speed, distribution particle size, screening efficiency and loss rate obtained through simulation are not enough. The interaction process between particles and particles-sieve on the sieve surface are lacking, and these can just reflect the sieving process of sieving device.
Response: Thanks for your comments. In the early stages of the study, we analyzed the movement characteristics of individual particles on the sieve surface and we obtained the following:
The three equations above are concerned with the analysis of the movement characteristics of individual particles on the sieve surface. We have not included a theoretical analysis of single particle motion in this paper, considering that individual particles produce different states of motion on the sieve surface and that the large number of particles colliding with each other on the sieve surface leads to a high degree of randomness in particle motion, making it more difficult to analyze the sieving process of particle populations through theory. We have now added this section to the text in its entirety.
Comment 2: The manuscript lacks the theoretical analysis of the sieving process of sieving device and the determination of the experimental factors. Author does not explain why the screen amplitude (SA), vibration frequency (VF), inclination angle (IA) and their levels are selected as the experimental factors, which makes the analysis of the effect of the experimental factors not deep enough.
Response: Thank you for your careful work. The motion of the oil soya bean is carried out under the action of the excitation force on the sieve surface. In the previously published research results we have established the equation for the velocity and acceleration at any point on the sieve surface as follows:
Through analysis and a study of the relevant literature, we have chosen screen amplitude (SA), vibration frequency (VF), inclination angle (IA) as the test factors. We have cited the relevant references in the text.
Comment 3: The test plan of the manuscript is too simple and needs to be optimized. Does the interaction of the screen amplitude, vibration frequency and inclination angle affect the test evaluation index?
Response: The interaction of the test factors influenced the test evaluation indicators and the main test results are shown in the table below, we have modified the specifics in the text.
Table 1. Codes of experimental factors.
|
Codes |
Factors |
||
|
X1 (Hz) Vibration frequency |
X2 (mm) Screening amplitude |
X3 (°) Inclination angle |
|
|
-1 |
14 |
6 |
2 |
|
-0.5 |
16 |
8 |
4 |
|
0 |
18 |
10 |
6 |
|
0.5 |
20 |
12 |
8 |
|
1 |
22 |
14 |
10 |
Table 2. Codes of experimental factors
|
No. |
Factors |
Evaluation index |
|||
|
X1 (Hz) Vibration frequency |
X2 (mm) Screening amplitude |
X3 (°) Inclination angle |
Y1 (%) Screening efficiency rate |
Y2 (%) Loss rate |
|
|
1 |
-1 |
-1 |
-1 |
88.69 |
0.18 |
|
2 |
-1 |
-0.5 |
-0.5 |
91.53 |
0.61 |
|
3 |
-1 |
0 |
0 |
95.30 |
2.79 |
|
4 |
-1 |
0.5 |
0.5 |
96.92 |
4.06 |
|
5 |
-1 |
1 |
1 |
96.73 |
4.42 |
|
6 |
-0.5 |
-1 |
-0.5 |
89.15 |
0.59 |
|
7 |
-0.5 |
-0.5 |
0 |
93.80 |
1.97 |
|
8 |
-0.5 |
0 |
0.5 |
96.72 |
3.28 |
|
9 |
-0.5 |
0.5 |
1 |
97.15 |
3.88 |
|
10 |
-0.5 |
1 |
-1 |
96.08 |
3.92 |
|
11 |
0 |
-1 |
0 |
91.89 |
1.71 |
|
12 |
0 |
-0.5 |
0.5 |
95.70 |
2.94 |
|
13 |
0 |
0 |
1 |
97.37 |
3.94 |
|
14 |
0 |
0.5 |
-1 |
95.80 |
4.23 |
|
15 |
0 |
1 |
-0.5 |
96.69 |
6.17 |
|
16 |
0.5 |
-1 |
0.5 |
94.69 |
1.90 |
|
17 |
0.5 |
-0.5 |
1 |
97.13 |
3.32 |
|
18 |
0.5 |
0 |
-1 |
95.47 |
3.75 |
|
19 |
0.5 |
0.5 |
-0.5 |
96.60 |
6.03 |
|
20 |
0.5 |
1 |
0 |
96.26 |
7.30 |
|
21 |
1 |
-1 |
1 |
96.02 |
3.44 |
|
22 |
1 |
-0.5 |
-1 |
96.03 |
2.80 |
|
23 |
1 |
0 |
-0.5 |
96.56 |
5.51 |
|
24 |
1 |
0.5 |
0 |
97.18 |
7.26 |
|
25 |
1 |
1 |
0.5 |
96.79 |
8.09 |
|
26 |
0 |
0 |
0 |
94.24 |
3.31 |
|
27 |
0 |
0 |
0 |
94.65 |
2.95 |
Table 3. ANOVA table of screening efficiency
|
Sources of variance |
Sum of squares |
Degrees of freedom |
Mean variance |
F value |
P value |
|
Model |
140.64 |
9 |
15.63 |
64.07 |
< 0.0001** |
|
X1 |
2.11 |
1 |
2.11 |
8.64 |
0.0088** |
|
X2 |
15.01 |
1 |
15.01 |
61.55 |
< 0.0001** |
|
X3 |
20.34 |
1 |
20.34 |
83.40 |
< 0.0001** |
|
X1X2 |
1.64 |
1 |
1.64 |
6.71 |
0.0185* |
|
X1X3 |
6.58 |
1 |
6.58 |
26.99 |
< 0.0001** |
|
X2X3 |
9.75 |
1 |
9.75 |
39.99 |
< 0.0001** |
|
X12 |
5.39 |
1 |
5.39 |
22.09 |
0.0002** |
|
X22 |
1.24 |
1 |
1.24 |
5.09 |
0.0367* |
|
X32 |
5.61 |
1 |
5.61 |
23.01 |
0.0001** |
|
Residual |
4.39 |
18 |
0.24 |
|
|
|
Lack of fit |
4.07 |
16 |
0.25 |
1.59 |
0.4542 |
|
Pure error |
0.32 |
2 |
0.16 |
|
|
|
Total |
145.03 |
27 |
|
|
|
Table 4. ANOVA table of loss rate
|
Sources of variance |
Sum of squares |
Degrees of freedom |
Mean variance |
F value |
P value |
|
Model |
103.51 |
9 |
11.50 |
82.97 |
< 0.0001** |
|
X1 |
11.06 |
1 |
11.06 |
79.82 |
< 0.0001** |
|
X2 |
29.58 |
1 |
29.58 |
213.36 |
< 0.0001** |
|
X3 |
3.65 |
1 |
3.65 |
26.30 |
< 0.0001** |
|
X1X2 |
1.82 |
1 |
1.82 |
13.12 |
0.0019** |
|
X1X3 |
0.82 |
1 |
0.82 |
5.91 |
0.0257* |
|
X2X3 |
1.12 |
1 |
1.12 |
8.07 |
0.0108* |
|
X12 |
1.85 |
1 |
1.85 |
13.33 |
0.0018** |
|
X22 |
0.61 |
1 |
0.61 |
4.41 |
0.0501 |
|
X32 |
0.016 |
1 |
0.016 |
0.11 |
0.7398 |
|
Residual |
2.50 |
18 |
0.14 |
|
|
|
Lack of fit |
2.42 |
16 |
0.15 |
3.82 |
0.2272 |
|
Pure error |
0.079 |
2 |
0.040 |
|
|
|
Total |
106.00 |
27 |
|
|
|
Comment 4: The working parameters of the sieve surface in the field test and its realization method are not specified. There are too few field verification experiments. The optimal parameters in the simulation test cannot be proved to be the optimal parameters in practical application; and it is necessary to carry out the adaptability test of different varieties of tiger nut.
Response: Thanks for your comments. The field trials in the text are based on the analysis at that time as we were unable to complete the orthogonal trials at the time of the first field trial. The results of the orthogonal tests were then verified in a further field trial. In this case, the working parameters were adjusted by changing the gearing and the length of the screen linkage.
Comment 5: The discussion part of the manuscript does not meet the writing requirements of this part.
Response: Thanks for your comments. This section has been modified in the paper.
Comment 6: The manuscript lacks relevant pictures of the simulation process. It is recommended to combine these pictures to explain the impact analysis of experimental factors.
Response: Thank you for your careful work. We have included a vector of particle movement during the sieving process so that the randomness of the particle population movement can be more clearly observed and the sieving process of oleaginous beans can be analyzed so that the content of the article can be more relevant to the topic.
Comment 7: The data in the manuscript discussion is inconsistent with the previous text, which makes the data in the manuscript questionable.
Response: Thank you for your careful work. We have rechecked all the data in the text to make sure that the data are consistent.
Comment 8: Some characters and numbers are wrongly written, such as Shear modulus of Steel. “As can be seen from Figure 8,” should be “As can be seen from Figure”.
Response: Thank you for your careful work. Changes have been made to this section.
Comment 9: The relevant expressions in the manuscript should be consistent, such as tiger nut particles, oilseed beans, and oleaginous beans should be unified.
Response: Thank you for your careful work. Relevant terminology has been revised.
Comment 10: The references are almost entirely in Chinese, which is very bad.
Response: Thank you for your careful work. We have added some new references.
Comment 11: Some references are formatted incorrectly, Shun as NO. 23.
Response: Thank you for your careful work. This section has been amended in the text and other similar issues have been checked.
